# Quality by Design (QbD) Based Method for Estimation of Xanthohumol in Bulk and Solid Lipid Nanoparticles and Validation

**DOI:** 10.3390/molecules28020472

**Published:** 2023-01-04

**Authors:** Vancha Harish, Waleed Hassan Almalki, Ahmed Alshehri, Abdulaziz Alzahrani, Madan Mohan Gupta, Sami I. Alzarea, Imran Kazmi, Monica Gulati, Devesh Tewari, Gaurav Gupta, Kamal Dua, Sachin Kumar Singh

**Affiliations:** 1School of Pharmaceutical Sciences, Lovely Professional University, Phagwara 144411, India; 2Department of Pharmacology, Umm Alqura College of Pharmacy, Umm Alqura University, Makkah 21955, Saudi Arabia; 3Department of Pharmacology & Toxicology, College of Clinical Pharmacy, Imam Abdulrahman Bin Faisal University, King Faisal Road, P.O. Box 1982, Dammam 31441, Saudi Arabia; 4College of Clinical Pharmacy, Pharmaceutical Chemistry Department, AlBaha University, P.O. Box 1988, Al Baha 65411-14 KSA, Saudi Arabia; 5School of Pharmacy, Faculty of Medical Sciences, The University of the West Indies, St. Augustine 3303, Trinidad and Tobago; 6Department of Pharmacology, College of Pharmacy, Jouf University, Sakaka 72341, Saudi Arabia; 7Department of Biochemistry, Faculty of Science, King Abdulaziz University, Jeddah 80203, Saudi Arabia; 8Faculty of Health, Australian Research Centre in Complementary & Integrative Medicine, University of Technology Sydney, Ultimo, NSW 2007, Australia; 9Department of Pharmacognosy and Phytochemistry, School of Pharmaceutical Sciences, Delhi Pharmaceutical Sciences and Research University, New Delhi 110017, India; 10School of Pharmacy, Suresh Gyan Vihar University, Mahal Road, Jagatpura, Jaipur 302017, India; 11Department of Pharmacology, Saveetha Institute of Medical and Technical Sciences, Saveetha Dental College and Hospitals, Saveetha University, Chennai 600077, India; 12Uttaranchal Institute of Pharmaceutical Sciences, Uttaranchal University, Dehradun 248007, India; 13Discipline of Pharmacy, Graduate School of Health, University of Technology Sydney, Ultimo, NSW 2007, Australia

**Keywords:** RP-HPLC, anticancer, BBD, beer, AQbD, humulus, validation

## Abstract

The analytical quality by design (AQbD) approach is utilized for developing and validating the simple, sensitive, cost-effective reverse-phase high performance liquid chromatographic method for the estimation of xanthohumol (XH) in bulk and nanoformulations. The Box–Behnken design (BBD) is applied for method optimization. The mobile phase ratio, pH and flow rate were selected as independent variables, whereas retention time, peak area, peak height, tailing factor, and theoretical plates were selected as dependent variables. The chromatogram of XH obtained under optimized conditions has given optimum conditions such as retention time (5.392 min), peak area (1,226,737 mAU), peak height (90,121 AU), tailing factor (0.991) and theoretical plates (4446.667), which are contoured in the predicted values shown by BBD. Validation of the method has been performed according to ICH Q2(R1) recommendations, using optimized conditions for linearity, limit of detection (LOD) and limit of quantification (LOQ), accuracy, precision, robustness and system suitability. All the values of validation parameters lie within the acceptable limits prescribed by ICH. Therefore, the developed and validated method of XH by the AQbD approach can be applied for the estimation of XH in bulk and various nanoformulations.

## 1. Introduction

Xanthohumol (XH) is the most abundant prenylated chalcone isolated from the feminine inflorescences of beer hops (*Humulus lupulus* L). It is considered to be one of the most well characterized prenyl flavonoid derivatives discovered to date. It has been reported to exhibit a plethora of pharmacological effects [1]. It is also used as the main flavoring ingredient in beer at low concentrations and also as a food supplement [2]. Due to its anti-oxidant property, it can help in the treatment of diseases such as cancer that are caused by oxidative stress. XH has proven to prevent the growth of a variety of human cancers, such as ovarian cancer, breast cancer, prostate cancer, colon cancer and leukemia [3].

Owing to its high medicinal value and anticancer potential [2,4], XH has been developed as a novel molecule of interest for research. Chemically, XH is 1-(2,4-dihydroxy-6-methoxy-3-(3-methyl-2-butenyl) phenyl)-3-(4-hydroxyphenyl)-2-propen-1-one (Figure 1) with a molecular weight of 354.4 g/mol [5]. XH has the capacity to inhibit carcinogenesis at all stages, thus being classified as a broad spectrum anticarcinogenic agent. It also functions as an anti-initiator by inhibiting the activity of procarcinogen activation enzymes (CYP450) and detoxifying enzymes of cancer (glutathione-s-transferase) [6].

To harness the therapeutic potential of XH and its clinical translation, its suitable dosage form needs to be developed. Some products of XH are available on the market as nutraceuticals in the form of capsules such as xanthoforce hop extract, super smart xanthohumol and curcumin and the xanthohumol dietary supplement. However, there are still many more options of nutraceutical compositions of XH to be developed. Analytical method development plays a major role in formulation development, especially during the early pre-formulation studies and characterization of various biopharmaceutical parameters such as dissolution, permeation and pharmacokinetics. One of the bottlenecks behind the low popularity of XH as a nutraceutical is the lack of an analytical method to support its formulation development and quantitation of XH specifically from the extracts. Another challenge associated with XH is its lipophilicity and high cytosolic protein binding. Hence, the development of a novel drug delivery system to overcome its dissolution rate and cytosolic binding of limited oral bioavailability seems to be an attractive strategy to overcome this limitation.

The accurate, precise, sensitive and specific analytical method further helps in the assessment of drug loaded in a nanoparticle in trace amounts either from the dosage form or during its metabolism and pharmacokinetic studies. To date, there are only three LC-MS/MS methods available for the detection of XH. In one of the studies, Vázquez Loureiro et al. [7] determined the amount of XH in food supplements, beers and hops by using HPLC-DAD. In another study, Sus et al. [8] developed a reversed phase liquid chromatographic technique for the quantification of prenylflavonoids and chalcones (8- prenylnaringenin (8-PN), isoxanthohumol (IXH), XH, 6-prenylnaringenin (6-PN), in human plasma and urine. Avula et al. [9] also developed the HPLC method to detect the XH in biological samples of rat plasma, urine and feces.

All of these methods are expensive and used exclusively for the estimation of XH in biological fluids. None of the reported methods have shown their applicability for use in terms of a routine quality control purpose. In addition, the reported methods have utilized a traditional method development approach using one variable at a time (OVAT). OVAT consumes more time and it is difficult to understand the critical parameters. Analytical quality by design (AQbD) aids in the complete understanding of conceivable risk and concomitant interactions among the variables of the method, based on quality risk management and concepts of design experiments. It mainly focuses on the risk-based critical parameters that affect the performance of the method to produce a novel, robust analytical method. Liquid chromatography has evolved over the years as a preferred inexpensive and nearly indispensable instrument for drug analysis. RP-HPLC has lately contributed significantly to the progress of analytical research, with applications in nutraceuticals, specialized metabolites, pharmaceuticals, plastics, foods, environmental monitoring, clinical research and polymers [10,11]. A simple, accurate, sensitive and efficient technique was established in this study using AQbD to detect XH in bulk and solid lipid nanoparticles (SLNs). The advantage of using the AQbD approach over OVAT includes rapid rate, simultaneous detection capacity, high sensitivity and automated operation. Box–Behnken Design (BBD) was selected for the current study, as it is a three factor, three level design that is preferred over other designs such as CCD and Doehlert design because it necessitates fewer experimental runs, is rotatable, and does not simultaneously contain the highest or lowest levels of the cubic region; as a result, the likelihood of receiving substandard results is reduced. BBD also makes it possible for sequential evaluation of various factors by avoiding the criteria of iso-variance. Therefore, BBD is selected for developing the HPLC analytical method to estimate XH [12,13].

## 2. Results and Discussion

### 2.1. Preliminary HPLC Method Development Studies

The RP-HPLC method for the estimation of XH in bulk and nanoformulation was developed and optimized by the AQbD approach. To develop the method, preliminary investigations were carried out in compliance with protocols published in the literature and pharmacopoeias [14,15,16]. Many attempts were made with varying compositions of the mobile phase containing solvents (acetonitrile, methanol), and OPA 0.1% *v*/*v*, pH of 0.1% *v*/*v* OPA (1.5, 1.7, and 1.9), flow rate (0.8, 1.0, and 1.2 mL/min) and column oven temperature (26–40 °C), to improve the efficiency of the method and for the least peak tailing and better peak resolution. Several attempts have failed to give satisfactory peak resolution, tailing factor, theoretical plates and less retention time. The mixture of 0.1% *v*/*v* OPA and acetonitrile did not result in a peak. Finally, methanol and 0.1% *v*/*v* OPA (75:25% *v*/*v*) was used, which resulted in a peak with optimum resolution, retention time, theoretical plates and peak tailing. The chromatographic conditions for the developed robust RP-HPLC method for XH were optimized by BBD.

### 2.2. Risk Assessment Studies

According to the principle of quality risk management, risk assessment studies are deemed necessary for identifying numerous causes and sub-causes of variability, as well as their probable relationships with selected critical analytical attributes. Table 1 depicts the results of various critical parameters studied for risk assessment. These assessed parameters show the connection between diverse analytical attributes and parameters of method. The criticality and probability of numerous input elements were evaluated based on the study of several relevant publications and distinct degrees of risk connected with individual method parameters. These were described in the risk estimate matrix.

Among all the parameters, mobile phase ratio, pH of the 0.1% *v*/*v* OPA and flow rate were considered to be having high risks. On the other hand, column dimension was found to be in medium risk, and all other parameters such as injection volume, column temperature, and flow type were at low risk assessment and are expected to exert a insignificant effect on the performance of the method. A controlled approach for continuous improvement was established based on the best outcome achieved through screening and risk assessment, method optimization by employing experimental designs, modeling and optimal search using response surface methods to enter the analytical design space. Therefore, the risk was minimized after optimization by applying the experimental design. In addition, risk based assessments simplified the anticipation of vulnerability and its severity to the method as a function of input parameters.

### 2.3. Optimization

BBD was applied for the optimization of the chromatographic conditions. The variables employed for the separation of XH in bulk were A: mobile phase ratio (Methanol: 0.1% *v*/*v* OPA), B: pH (1.5) of 0.1% *v*/*v* OPA and C: flow rate at low (−1), medium (0) and high (+1) levels. To get the best HPLC conditions, the design included seventeen trial runs with five center points (Table 2). These 17 experiments were performed based on the design and the responses checked against these variables were retention time (R1), peak area (R2), peak height (R3), tailing factor (R4), and theoretical plates (R5). The design has given the optimized chromatographic conditions with the mobile phase ratio (10:90 % *v*/*v*), pH (1.9 pH) and flow rate (0.8 mL/min) with a desirability of 0.788, which is near to 1.

All the critical parameters observed are found within the target limits. In addition, a graphical optimization was carried out, with the best option being identified within the design space. The HPLC analysis was performed by utilizing the best chromatographic conditions. Using the optimized method, the chromatogram of the XH was viewed. The effects of the factors (independent variables) on the responses (dependent variables) such as retention time, chromatogram area, tailing factor and theoretical plates were studied. The responses obtained from various runs of the experiment were fitted into various kinetic models such as linear, second order (2FI), quadratic and cubic models.

All of the tested independent variables were studied for the kinetic models. The results showed the quadratic model as the best fit for retention time and peak area as their *p* values were found to be less than 0.0001. The other variables i.e., peak height, tailing factor and theoretical plate showed best fit in linear models as their *p* values were less than 0.05. The independent variables kinetic (quadratic and linear) results showed the highest r2 value of 0.9970 which is quite close to 1. These results have confirmed that all the responses (independent variables) showed combined and individual effects on the dependent variables (Table 3). For every response, contour and 3D plot was prepared and evaluated to observe the effects of factors on responses (Figure 2).

### 2.4. Effects of Independent Variables on Dependent Variables

#### 2.4.1. Effect of Independent Variables on Retention Time

The F-value of the model was found to be 262.02 with a *p*-value < 0.05 and the lack of fit F-value was identified as 800.35. This indicates that the model was significant. There was only a 0.01% chance that model F-value and lack of fit F-value could occur due to the noise (Table 2). The polynomial equation, 3D plots and contour plots have shown the relationship among the mobile phase ratio (A), pH of 0.1% *v*/*v* OPA(B) and flow rate (C) on retention time (Figure 2). The ratio of mobile phase has exhibited a substantial effect on the retention time of XH. As the concentration of the mobile phase (methanol) ratio was increased, the retention time of XH was significantly decreased while a decrease in concentration of the mobile phase increased the retention time of XH drastically. Furthermore, middle concentration of the mobile phase has also shown higher retention time.

It is a well-known fact that the method producing higher retention time is not desirable for developing the analytical method, prolonging the run time of the chromatogram and thereby leading to unnecessary wastage of the mobile phase, as well as time [17]. Therefore, the highest ratio was selected for the development of the analytical method for XH. The second factor i.e., pH (1.5) of 0.1% *v*/*v* OPA did not have a significant effect on the retention time. As the pH increased or decreased, there was a non-significant change in the retention time which indicated the absence of any effect on the retention time. Flow rate (C) is the third factor which has a significant effect on the retention time. As the flow rate was increased, the retention time decreased, and vice versa. The combined effect of the mobile phase ratio and pH of 0.1% *v*/*v* OPA showed a positive effect on retention time of XH. The combined effects of factors i.e., flow rate (C) and the mobile phase ratio (A) showed positive effect on the retention time. The other factors, flow rate (C) and pH (B) showed a positive effect on the retention time.

The polynomial equation has clearly shown that the factors A (ratio of mobile phase), B (pH of 0.1% *v*/*v* OPA), and C (flow rate) had a negative effect individually and a positive effect in combination on the retention time of XH. Therefore, it was concluded that on increasing the levels of A, B and C factors there was a decrease in the retention time of XH. All the factors A, B, C, AB, AC, BC, A2, B2, and C2 were found to be significant model terms as shown in the polynomial equation. The quadratic polynomial equation is shown in Equation (4).
Retention time (R1) = +6.05 − 3.36 × A − 0.0710 × B − 1.40 × C + 0.2185 × AB + 0.7228 × AC + 0.0750 × BC + 1.37 × A^2^ + 0.0665 × B^2^ + 0.2163 × C^2^
(1)

#### 2.4.2. Effect of Independent Variables on Peak Area

The F-value of the model was found to be 290.50 with a *p*-value < 0.05, which indicated that the model was significant. The Lack of Fit F-value was identified as 94.28, which indicated that the lack of fit was significant. There was only a 0.01% chance that model F-value and a 0.04% chance that Lack of Fit F-value could occur due to the noise (Table 2). The polynomial equation, 3D plots and contour plots have showed the relationship between the mobile phase ratio (A), pH of 0.1% *v*/*v* OPA(B) and flow rate (C) on peak area (Figure 2B). The positive effect of both mobile phase ratio (A) and pH(B) was observed on the peak area. At the initial stage, a slight increase in the factors A and B showed small deviation in the peak area. Later on, an increase in the mobile phase ratio and pH significantly increased the peak area.

The other factor flow rate (C) has a synergistic negative effect on the peak area. The increase in flow rate from 0.8 mL/min to 1.2 mL/min caused a significant decrease in the peak area. The combined effect of mobile phase ratio (A) and flow rate (C) has a significant negative effect on the peak area. At high levels of the mobile phase ratio and flow rate, the peak area has significantly decreased and at low levels, the peak area has increased. Hence, it was confirmed that the higher levels of flow rate and the mobile phase ratio would result in a sharp peak with good area. The independent variables, i.e., ratio of mobile phase (A) and pH (B) also had negative effect on the peak area. However, the combined effect of factors pH (B) and flow rate (C) has a positive effect on the peak area. All the factors A, B, C, AB, AC, BC, A2, B2, and C2 were considered as significant model terms, as shown in the polynomial equation. The quadratic polynomial equation is shown in Equation (5).
Peak area (R2) = +9.746 × 10^5^ + 4317.50 × A + 19,929.00 × B − 2.075 × 10^5^ × C − 3036.50 × AB − 2771.00 × AC + 14,296.00 × BC − 5200.10 × A^2^ − 15,703.10 × B^2^ + 28,911.40 × C^2^(2)

#### 2.4.3. Effect of Independent Variables on Peak Height

The F-value of the model was found to be 8.14 with a *p*-value less than 0.05, indicating that the model was significant. The Lack of Fit F-value was identified as 15,4817.5, which indicated that the lack of fit was significant. There was only a 0.26% chance that the model F-value and 0.01% chance that Lack of Fit F-value could occur due to the noise (Table 2). The linear polynomial equation has shown the connection between mobile phase ratio (A), pH of 0.1% *v*/*v* OPA(B) and flow rate (C) on the peak height (Figure 3A). The effects of factors, mobile phase ratio(A), pH of 0.1% *v*/*v* OPA and flow rate (C) has been extensively studied on peak height. The design polynomial equation, contour and 3D plots have showed the collaborative positive effect of an independent variable such as mobile phase ratio (A) and pH of 0.1% *v*/*v* OPA and a negative effect of variable C (flow rate) on peak height. An increase in the ratio of the mobile phase leads to an increase in the peak height of XH. The pH of 0.1% *v*/*v* OPA has a positive effect on peak height but there is a slight change in the height of the peak by increasing the pH (1.5) of OPA (0.1% *v*/*v*). The third factor flow rate (C) also shows an impact on the peak height, as the flow rate increased from 0.8 mL/min to 1.2 mL/min the peak height of XH was decreased significantly. This implied that the lower flow rate would result in maximum height of the peak. All the factors A, B and C were found to be significant model terms, as shown in the linear polynomial equation (Equation (6)).

The Linear polynomial equation is as follows:Peak height (R3) = +60,279.24 + 23,412.87 × A + 2082.13 × B − 94,95.00 × C (3)

#### 2.4.4. Effect of Independent Variables on Tailing Factor

The F-value of the model was found to be 27.72 with a *p*-value less than 0.05, which indicated that the model was significant. The Lack of Fit F-value was identified as 0.19, which indicated that the lack of fit was significant (Table 2). The relationship between the mobile phase ratio (A), pH of 0.1% *v*/*v* OPA(B) and flow rate (C) on the tailing factor has been explained by polynomial equation, 3D plots and 2D-contour plots (Figure 3B). It was observed that mobile phase ratio (A) has a significant effect on tailing factor. An increase in the level of the mobile phase ratio caused a significant decrease in the tailing factor. The pH (1.5) of 0.1% *v*/*v* OPA had a non-significant effect on tailing factor. An increase in the pH caused a slight decrease in the tailing factor of XH’s chromatogram. Another factor, i.e., flow rate (C) has a positive effect on tailing factor; however, the impact on tailing of the peak was minimal, as a higher tailing factor is not recommended for the HPLC method [17]. Hence, the ratios of factors A, B and C were found to be in an accepted limit (<2) and have been chosen for the method development. The polynomial equation clearly showed that there is only an individual effect of factors on the tailing factor, but there is no combined effect. All the factors A, B and C were found to be significant terms in the model. The polynomial equation of the linear model of tailing factor (R4) is shown in Equation (7).
Tailing factor (R4) = +0.8449 − 0.091 × A − 0.0040 × B + 0.0072 × C(4)

#### 2.4.5. Effect Independent Variables on Theoretical Plates

The F-value of the model was found to be 4.36 with a *p*-value less than 0.05. This indicated that the model was significant. The Lack of Fit F-value was identified as 2.59, which indicated that the lack of fit was significant (Table 2). The 2D-contour plot, 3D plots and polynomial equation showed the linear relationship between factors and theoretical plates (Figure 3C). The results have shown that with an increase in the ratio of mobile phase (A), the theoretical plates have increased in a linear pattern. Hence, a higher ratio of mobile phase has been chosen for method development. The factor pH of 0.1% *v*/*v* OPA also had a positive effect on the theoretical plate. It was observed that with an increase in the pH of OPA, the plate count also increased. Due to this, the highest pH was selected for the method. The flow rate has a negative effect on the theoretical plates and an increase in the flow rate from 0.8 to1.2 mL/min caused a gradual decrease in the theoretical plate count, which is not recommended. Hence, a lower level of flow rate has been selected for the method. The linear polynomial equation for theoretical plates is shown in Equation (8).
Theoretical plate (R5) = +3598.66 + 418.53 × A + 121.95 × B − 254.62 × C(5)

### 2.5. Optimization of Chromatographic Method

The numerical optimization method was used to identify the best chromatographic condition. The numerical optimization mainly involves the changing of various independent variables in order to achieve the desired conditions such as maximum peak area and theoretical plates, as well as minimum retention time and tailing factor, in order to achieve a desirability function value nearer to 1 [18] The optimized solution was obtained from the higher levels of the mobile phase ratio (10:90% *v*/*v*; methanol: 0.1% *v*/*v* OPA) as well as pH of 0.1% *v*/*v* OPA (1.9) and a lower level of flow rate (0.8 mL/min). The predicted solutions of optimized chromatographic conditions were retention time of 5.085 min, peak area of 1,199,750.526 mAU, peak height of 95,269.132 AU, tailing factor of 0.743 and a theoretical plate count of 4393.762 cm with a desirability value of 0.788. Along with the improved solution, graphical optimization was used to demarcate the best analytical design space area (Figure 4 and Figure 5).

The obtained value of retention time, peak area, peak height, tailing factor and theoretical plate were found to be 5.392 min, 1,226,737 mAU, 90,121 AU, 0.991 and 4446.667 cm, respectively. The *p* value between predicted values and obtained values for all the parameters were found to be more than 0.05, indicating non-significant variation and thus validating the experimental design for having excellent correlation between predicted and actual values. Thus, the adaption of a systematic AQbD method to achieve the best possible analytical solution under competing objectives might be credited to such perfect chromatographic results. It not only assisted in achieving the best analytical results but also revealed the underlying factor response connection. The typical HPLC chromatogram of XH obtained through the optimized method is shown in Figure 6A.

### 2.6. Method Validation

#### 2.6.1. Linearity

The results of a calibration curve (Figure 6B) deciphered that XH has obeyed good linearity with the working standard solutions in the range of 2–10 μg/mL. The calibration curve was found to be linear, with a decent regression coefficient (r2) of 0.9996.

#### 2.6.2. LOD and LOQ

The method has extremely low LOD and LOQ values of 0.425 μg/mL and 1.289 μg/mL, respectively, suggesting that the proposed method for XH estimation has a high sensitivity.

#### 2.6.3. Accuracy (Recovery Method)

Accuracy of the method was performed by the recovery method; the recovery for LQC, MQC and HQC was appreciable and % recovery was found to be in between 101.1% to 109.7%. The % RSD values were less than two. Therefore, the results were found to be within limits, which indicated that the developed method for estimation of XH has accuracy of a high degree. The results of the accuracy study are shown in Table 4.

#### 2.6.4. Precision

The precision of the method was determined by using three quality control standards (LQC, MQC and HQC), and the obtained results of the study are shown in Table 5. The % RSD of inter-day (0.64–1.10%), inter-analyst (0.594–0.889%) and intra-day (0.738–0.994%) precision was less than 2%, which indicated that the results were within the acceptable limits and the developed method was precise.

#### 2.6.5. Robustness

The robustness of the method was performed by changing the wavelength (365, 370 and 375 nm) and flow rate (0.6, 0.8, 1.0 mL/min). The results of % RSD were found to be less than 2%, which indicated that the method was robust. The results of robustness are shown in Table 6.

### 2.7. Parameters of System Suitability

After six injections of 10 μg/mL, the system suitability findings revealed that there was no significant change in the critical attributes, i.e., peak area, retention time, theoretical plates, peak purity index and peak tailing factor of XH. System suitability parameters are shown in Table 7.

#### System Specificity

The system specificity was assessed by injecting the blank samples of all the excipients used in the preparation of SLNs. No significant peak was observed at the retention time of pure XH. Hence, the developed method was found to be specific for XH. The chromatograms of all the excipients are shown in Figure 7.

### 2.8. Applications of Developed and Validated Method for Characterization of SLNs

#### Determination of % EE

The % EE of SLNs was determined by calculating the amount of entrapped and unentrapped XH in SLNs by the validated method. The data was collected in triplicate for three different batches of SLN with different compositions. The data of entrapped and unentrapped drug in SLNs is shown in Table 8. The data revealed a high level of repeatability with %RSD (<0.6) with % recovery 99.13 ± 0.14 to 100.9 ± 0.5 indicated that the method was specific and relevant to the assessment of the entrapment efficiency of XH in formulations based on nanocarriers.

In summary, the work demonstrated that AQbD concepts can successfully be used to develop an RP-HPLC method for XH. The ideal chromatographic conditions were determined using the strategy of describing the method objectives in the form of TMQP and identifying significant method qualities in the form of CAAs. Effective risk assessment studies, together with the appropriate use of experimental designs (response surface designs), made it simple to identify important method parameters as CAFs, which were then optimized to obtain the best chromatographic solutions to meet the method’s objectives.

Furthermore, the developed method’s efficiency was further verified by a validation methodology, allowing it to be used to estimate XH in both bulk and nanoformulations research. So far, there is no available RP-HPLC method developed by using AQbD approach for XH. The method developed is mainly used for the detection of XH and related compounds in the hop extracts and in beers but not specially for XH pure drug and nanoformulations. All the developed methods have a high retention time (>10 min) [7,8,9] which requires more time for a sample analysis. Therefore, the developed and validated RP-HPLC is an efficient method to estimate XH in bulk and nanoformulations with less retention time (5.392 min) while being cost-effective and robust.

## 3. Materials and Methods

### 3.1. Materials and Equipments Used

XH (Xantho Flav, 75%) was gifted by Simon H. Steiner, Hopfen, GmbH (Mainburg, Germany). Methanol and orthophosphoric acid (OPA) of HPLC grade were purchased from Molychem (Mumbai, India). Water was passed through a Milli Q filter to prepare the mobile phase. Compritol E ATO was gifted by Gattefosse, (Mumbai, India). Lipoid E80 SN was gifted by Lipoid GmbH (Germany), Pluronic F-68 was purchased from Hi-Media laboratories, (Mumbai, India). All other chemicals used were of an analytical grade.

The quantitative estimation of XH was done by using Shimadzu HPLC system (Japan) equipped with mobile phase delivery system (LC-20 AD), photodiode array detector (SPDM20A) and rheodyne sample injecting loop (20 μL). LC solution software was used as a data station for gathering information related to the chromatogram.

### 3.2. Chromatographic Conditions

The reverse phase (RP) HPLC method for quantification of XH was developed on RP C18 column (Nucleodur C18, 250 mm × 4.6 mm i.d., 5 μ) using 0.1% *v*/*v* OPA (pH 1.9) as aqueous phase and methanol as organic phase. Box–Behnken design (BBD) was used to optimize the chromatographic conditions such as mobile phase ratio, pH and flow rate through response surface methodology. The detection wavelength used was 370 nm and the sample injection volume was 20 μL.

### 3.3. Standards Preparation

#### Preparation of Stock and Working XH Solution

As the drug was completely soluble in methanol (99.5% *v*/*v*), it was used as a diluent to prepare stock and working solutions of XH. The primary stock solution of XH was prepared by dissolving 100 mg of drug in 100 mL of methanol in order to achieve the concentration of 1 mg/mL (1000 μL). The secondary stock solution was prepared by diluting 10 mL of XH primary stock solution to 100 mL using methanol to achieve XH concentration of 100 µg/mL. The working XH solution was prepared by diluting 10 mL of intermediate XH solution to 100 mL using methanol to achieve XH concentration of 10 μg/mL. Serial dilutions were made from the working XH solution so as to achieve the final concentrations (2, 4, 6, 8 μg/mL) using methanol.

### 3.4. Method Development

#### 3.4.1. Risk Assessment Studies

Risk assessment studies were used to interpret the impact of several factors impacting the target method quality profile (TMQP). Prior to the risk assessment, the critical analytical attributes were used to investigate formal relationships between critical method parameters of the TMQP (Table 9). It is used to classify the possible sources of a problem in order to figure out, as well as control the factors causing flaws, variations, defects, or failures [19]. During the analysis, the vital information is gathered from risk assessment studies, on the basis of risk and possibility connected to the distinct factors by assigning high, medium and low risk scores to the respective factor. Prioritization and screening of the factors was performed by picking seven factors and were separated on the basis of their risk scores. Out of these, three were chosen as critical analytical factors (CAF) for systematic optimization of critical analytical attributes (CAA’s) by utilizing experimental designs [20]. These CAF include composition of the mobile phase, pH and flow rate. To achieve an efficient method, CAAs relating to TMQP include peak area, retention time and tailing factor were determined [18,21,22]. The CAF and their scores with respect to CAAs are shown in Table 1 of results and discussion section (Section 2.2).

#### 3.4.2. Optimization of Chromatographic Method

The QbD approach is used to assess the impact of the insignificant value of independent parameters on dependent responses. It provides the necessary trial runs besides the interaction impact of covariates on dependent variables. Using the AQbD approach, this study was designed with three factors and three levels (+1, 0, −1) using Design Expert software version 11.1.0, Stat-ease Inc., Mineapolis, USA. Independent variables such as mobile phase (methanol: 0.1% OPA) (A % *v*/*v*), pH of 0.1% OPA (B; pH), flow rate (C; mL/min) and dependent variables such as retention time (R1, min), peak area (R2, mAU), peak height (R3, AU), tailing factor (R4), and theoretical plates (R5, cm) were used for the optimization of the method.

Peak area indicates the analyte concentration, whereas retention time of a drug is a measure of its ability to separate. Similarly, peak height indicates the intensity of the peak and tailing factor efficiency of the method. Theoretical plates indicate the mobile phase suitability and performance of the method. To obtain the optimal level, various factors used were combined into seventeen distinct experimental compositions using BBD as DOE tool. To assess the best fit model, the outcomes of all investigated conditions were fitted to quadratic, second order and linear equations with multiple regression technique. Furthermore, analysis of variance (ANOVA) was applied to understand the level of significance of each factor over the responses. The obtained polynomial equations have also helped in generating perturbation as well as 3D response surface plots. It is pertinent to add here that the method development was carried out using 10 μg/mL working standard solution of XH.

### 3.5. Method Validation

The validation of analytical techniques is important not only for regulatory objectives, but also for their long-term effectiveness and reliability in various research, quality control, process control, product development, clinical and toxicological investigations. This is significant because reliable target variability allows for the detection of atypical behavior of analytical data in routine applications such as quantitative analyte measurement, therefore increasing data quality reliability and reproducibility. It also proves that a certain analytical approach is dependable, predictable and repeatable. Therefore, the developed analytical method was validated for linearity, accuracy, precision, specificity, robustness, limit of detection (LOD) and limit of quantification (LOQ) as per ICH Q2 (R1) guidelines.

#### 3.5.1. Preparation of Quality Control Samples

Three different concentration levels of quality control samples were prepared, i.e., 4.8 μg/mL (LQC, lower quality control), 6 μg/mL (MQC, middle quality control) and 7.2 μg/mL (HQC, high quality control). The samples were stored at 4 °C for further analysis. The prepared samples were passed through a 0.22 μm syringe filter before proceeding to chromatography.

#### 3.5.2. Linearity

The linearity of the analytical method was evaluated with the prepared samples of concentration 2–10 μg/mL from the stock solution (1000 μg/mL) with n = 6. The linearity curve was plotted by taking peak area (mAU) on Y axis and concentration of XH (μg/mL) on X axis. Slope, regression equation and regression coefficient (r2) were calculated for linearity curve by using MS-Excel software [23].

#### 3.5.3. LOD and LOQ

The determination of LOD and LOQ for the method was performed at 3:1 and 10:1 signal to noise ratio by using standard deviation (σ) and slope of the standard curve [17]. LOD and LOQ were calculated using the formula given in Equation (1):LOD = 3.3*σ/S and LOQ = 10*σ/S(6)
where, σ is standard deviation and S is slope of the standard curve.

#### 3.5.4. Accuracy

Accuracy of the method was studied based on the absolute recovery of XH using its LQC, MQC and HQC concentrations that indicate recovery at 80%, 100% and 120%, respectively. The experiment was carried out in hexaplicate and the mean data, standard deviation (SD or σ), % relative standard deviation (% RSD) and % absolute recovery were calculated for each sample to confirm the accuracy data within specified limits [24]. The % absolute recovery was calculated using the formula given in Equation (2).
(7)$$\%\;\mathrm{Absolute}\;\mathrm{recovery}=\frac{\mathrm{Actual}\;\mathrm{concentration}\;\mathrm{recovered}}{\mathrm{Theoritical}\;\mathrm{concentration}}\times 100$$

#### 3.5.5. Precision

The consistency of results among several distinct aliquots of the identical concentration on the same day and other days of the analysis is explained by the precision of the developed analytical method. Intraday (repeatability), inter-day (reproducibility) and inter-analyst precision of the developed method was analyzed by injecting (six times) three diverse concentrations (LQC, MQC and HQC) of XH on initial day and three successive days under similar experimental conditions. The inter-analyst precision study was carried out by injecting hexaplicate samples of LQC, MQC and HQC by three different analysts on the same day. Mean data, SD, % RSD, recovery was computed [17].

#### 3.5.6. Robustness

Robustness is a measure of analytical technique consistency; it is a method’s ability to stay unchanged by modest purposeful alterations in parameters of the method. Robustness of the analytical method was analyzed by applying variations in flow rate (1.0, 0.8, 0.6 mL/min) and wave length (365, 370 and 375 nm). MQC (6 μg/mL) solution of XH was used for the experiment and its mean area, mean retention time and % RSD were calculated for analyzing the changes that took place in the chromatogram [25,26,27,28].

#### 3.5.7. System Suitability

System suitability testing is a method of ensuring a chromatographic system’s appropriateness for a specific analysis by verifying its resolution, column efficiency, and repeatability. A suitability test is predicted on the premise that the equipment, analytical procedures, electronics and samples to be analyzed are all part of a larger system that can be evaluated. This test was performed by injecting six injections of 10 μg/mL working solutions. %RSD, peak area, retention time, theoretical plates, tailing factor, peak purity index and height equivalent to theoretical plate (HETP) were determined and compared with their official limits [27,28].

### 3.6. Formulation of Solid Lipid Nanoparticles (SLNs)

SLNs of XH with different amounts of lipid and surfactant were prepared by homogenization, followed by probe sonication method. The desired quantity of lipid, drug, surfactant was weighed. Lipid phase and aqueous phase containing surfactant were prepared separately. The lipid phase was prepared by heating it above 10 °C of the lipid melting point, drug and lipid phase surfactant was added to the molten lipid. The aqueous phase was prepared by adding aqueous phase surfactant to water and heated to the temperature equal to that of the lipid phase (80 °C). Then, the aqueous phase was added to the lipid phase dropwise with continuous stirring using magnetic stirrer while maintaining its temperature. After complete addition of the aqueous phase to the lipid phase, the dispersion was homogenized for 30 min at 6000 rpm using a high speed homogenizer. Then, the dispersion was sonicated for 10 min at 40% amplitude and at a pulse rate of 30 using a probe sonicator. The resultant dispersion was cooled to room temperature so as to solidify the SLNs.

### 3.7. System Specificity

The purpose of the specificity assessment was to rule out any potential interaction between the drug and any of the excipients. All the excipients (Compritol E ATO, Lipoid E 80 SN and Pluronic F-68) used for preparing SLNs were diluted in ethanol/hexane according to their solubility profile and then injected to HPLC system after passing through 0.22 μm syringe filter. The interaction of the peak of drug with excipient peak is analyzed by comparing standard drug peak with the blank SLN peak [27].

### 3.8. Application of Developed and Validated Method for Characterization of SLNs

#### Determination of Percentage Entrapment Efficiency (% EE)

To determine the % EE, amount of entrapped and unentrapped drug has to be separated by passing through the sephadex G25 column. The drug present in the SLNs was extracted after lysis of lipid nanoparticles by mixing with methanol, followed by filtration through a 0.22 µm filter. Then, both entrapped and unentrapped XH content was determined in triplicate by the developed and validated method. The concentration of the XH was calculated by using the calibration curve of XH.

% EE was calculated by using the formula shown in Equation (3).
(8)$$\%\;\mathrm{EE}=\frac{\mathrm{Amount}\;\mathrm{of}\;\mathrm{drug}\;\mathrm{used}\;\mathrm{for}\;\mathrm{formulation}-\mathrm{amount}\;\mathrm{of}\;\mathrm{unentrapped}\;\mathrm{drug}}{\mathrm{Total}\;\mathrm{amount}\;\mathrm{of}\;\mathrm{drug}\;\mathrm{in}\;\mathrm{formulation}}\times 100$$

## 4. Conclusions

The simple, robust, cost-effective isocratic RP-HPLC method for the estimation of XH was successfully developed by applying the AQbD approach and validated according to the ICH Q2(R1) guidelines. The developed method was found to be more advantageous because the estimation time was less than 10 min per sample as compared to other methods where retention time was above 10 min. The factors were selected based on the risk assessment studies from the literature. The method development utilized BBD for optimizing the chromatographic conditions. The optimization was done by selecting mobile phase ratio (A), pH of 0.1% *v*/*v* OPA(B) and flow rate (C) as independent variables, whereas retention time (R1), peak area (R2), peak height (R3), tailing factor (R4), theoretical plates (R5) as dependent variables.

All the responses obtained from the experiments given by BBD were noted and graphical as well as numerical optimization was performed. The effects of all the independent variables on dependent variables have been studied. The optimized chromatographic conditions viz methanol:0.1% OPA; 10:90% *v*/*v*, pH;1.9 and flow rate 0.8 mL/min were used for performing further analysis-like detection of XH followed by validation of method. The well-defined peak was identified at retention time of 5. 392 min with peak area (1,226,737 mAU), peak height (90,121 AU), tailing factor (0.991) and theoretical plates (4446.667 cm). The linearity is performed with concentrations of range 2–10 μg/mL and a linear response (r2 = 0.9996) was observed. The sensitivity of the method was assessed by LOD and LOQ, which has given low values 0.425 μg/mL and 1.289 μg/mL indicating high sensitivity of the method. The accuracy study was carried out with three different concentrations (LCQ, MCQ and HCQ) and the % recoveries were found in the range of 101.1% to 109.7%. The interday, intraday and interanalyst precision values were found to be less than 2% which indicated the good precision of the method. The % RSD of robustness study was less than 2%, indicating the method is robust. The %RSD (<0.6) and % recovery 99.13 ± 0.14 to 100.9 ± 0.5 of specificity study indicated that the method was specific to XH. All the results obtained have been found to be within the acceptable range. Finally, the developed method can be used for the estimation of XH in bulk, as well as in various nano lipid-based carriers such as SLNs.

## Figures and Tables

**Figure 1 molecules-28-00472-f001:**
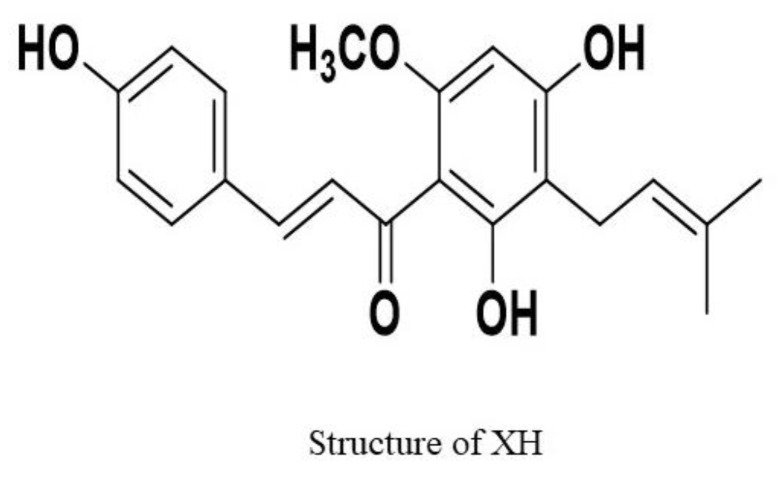
The chemical structure of XH.

**Figure 2 molecules-28-00472-f002:**
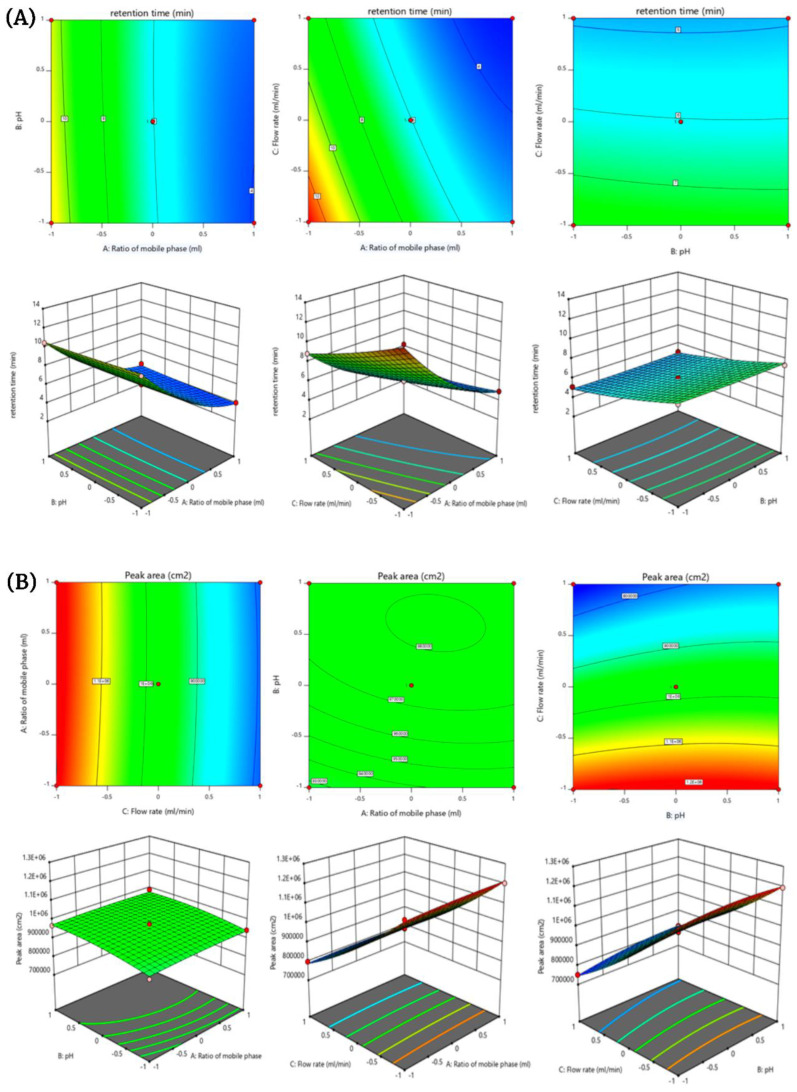
(**A**) 2D and 3D response surface plots representing the effect of factors Mobile phase ratio (A), pH of 0.1% OPA (B) and Flow rate (C) on retention time (R1) of XH. (**B**) 2D and 3D response surface plots representing the effect of factors Mobile phase ratio (A), pH of 0.1% OPA (B) and Flow rate (C) on Peak area (R2) of XH.

**Figure 3 molecules-28-00472-f003:**
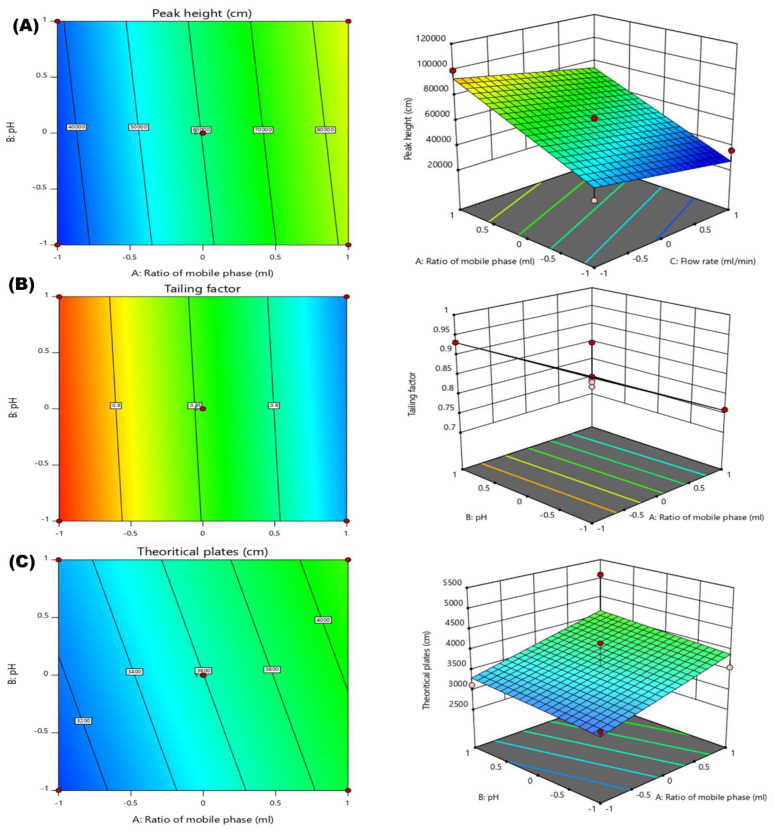
(**A**) 2D and 3D response surface plots representing the effect of factors Mobile phase ratio (A), pH of 0.1% OPA (B) and Flow rate (C) on Peak height(R3). (**B**) 2D and 3D response surface plots representing the effect of factors Mobile phase ratio (A), pH of 0.1% OPA (B) and Flow rate (C) on peak Tailing factor(R4). (**C**) 2D and 3D response surface plots representing the effect of factors Mobile phase ratio (A), pH of 0.1% OPA (B) and Flow rate (C) on Theoretical plates(R5).

**Figure 4 molecules-28-00472-f004:**
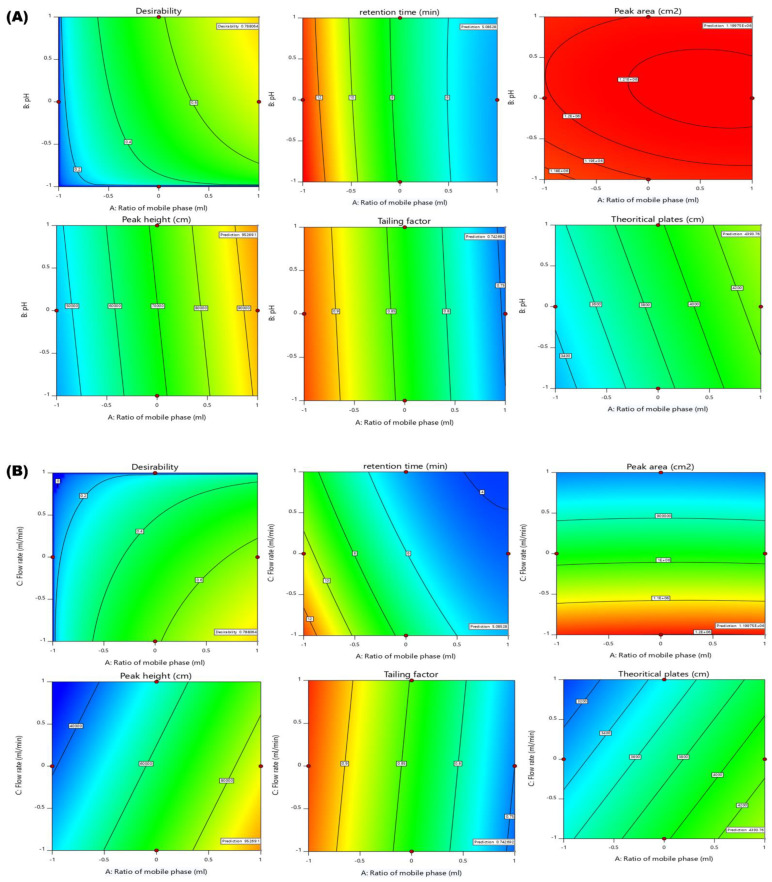
Optimized numerical 2D-Contour plots representing the optimal solutions and desirability of dependent variables; Retention time (R1), Peak area (R2), Peak height (R3), Tailing factor (R4), and Theoretical plates (R5). (**A**) Effect of the mobile phase ratio (A) and pH (B), (**B**) Effect of mobile phase ratio (A) and Flow rate (C), (**C**) pH (B), Flow rate (C).

**Figure 5 molecules-28-00472-f005:**
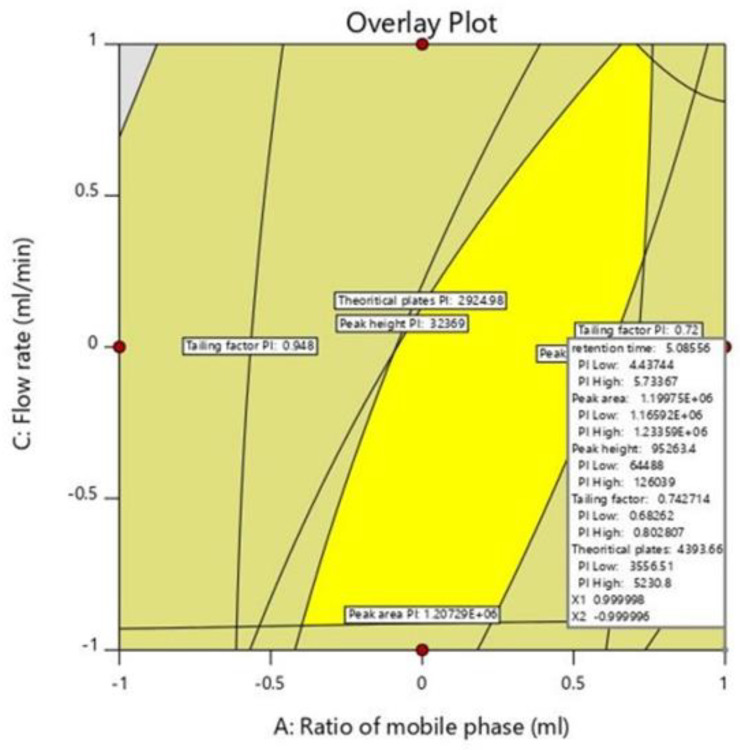
An overlay plot depicting the optimized method conditions for estimating XH.

**Figure 6 molecules-28-00472-f006:**
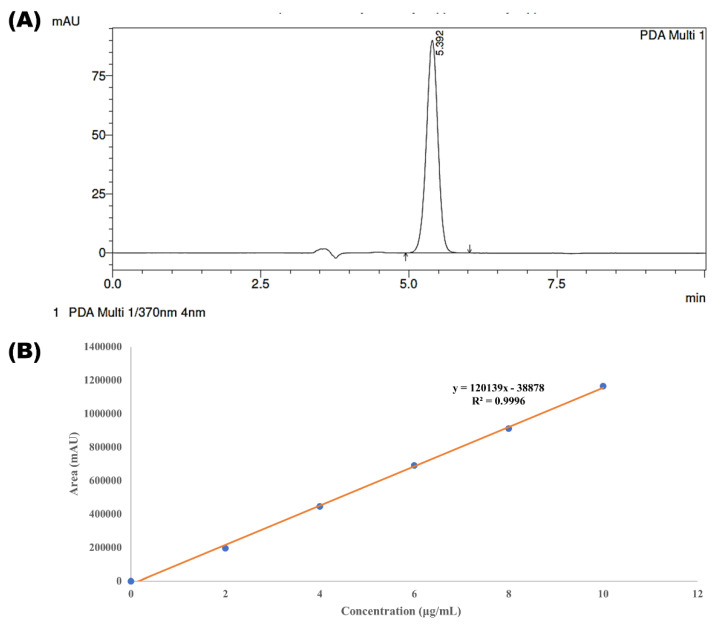
Depicting the chromatograms of (**A**) Xanthohumol and (**B**) calibration curve of XH

**Figure 7 molecules-28-00472-f007:**
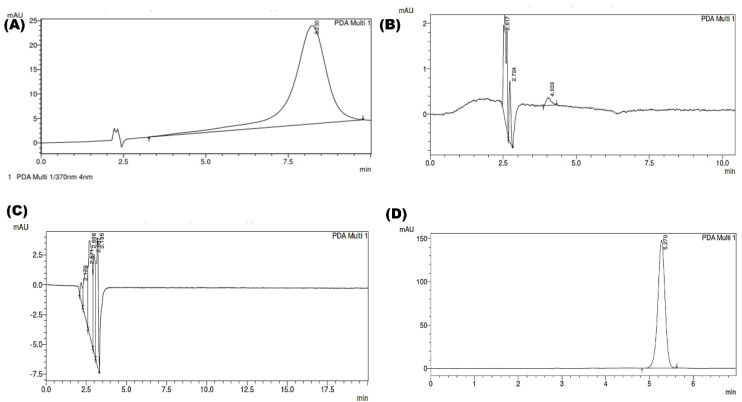
Depicting the chromatograms of (**A**) blank compritol E ATO (**B**) blank pluronic F 68 (**C**) blank lipoid S E80 (**D**) XH loaded SLN.

**Table 1 molecules-28-00472-t001:** Risk assessment parameters for developing the robust analytical method.

Chromatographic Method Parameters of XH							
CAAs	Composition of Mobile Phase	pH of 0.1% OPA	Column Temperature (20–40 °C)	Injection Volume	Flow Rate	Flow Type	Column Dimension
Peak area	+	+	−	−	+	−	0
Retention time	+	+	0	−	+	0	0
Peak height	+	+	−	−	+	0	0
Tailing factor	+	−	−	−	0	−	0

High risk (+), Medium risk (0), Low risk (−).

**Table 2 molecules-28-00472-t002:** Box–Behnken design representing the effects of independent variables on dependent variable at low, medium and high levels.

S.N.	Factor A: Ratio of Mobile Phase (mL)	Factor B: pH	Factor C: Flow Rate ml/min	Response (R1), Retention Time (min)	Response (R2), Peak Area (mAU)	Response (R3), Peak Height (AU)	Response (R4) Tailing Factor	Response (R5), Theoretical Plates (cm)
1	0	−1	1	5.129	751,181	57,142	0.840	3513.72
2	1	−1	0	4.045	944,677	92,949	0.760	3566.48
3	−1	1	0	10.477	968,790	35,711	0.932	3119.75
4	−1	0	1	8.856	799,071	36,086	0.948	2924.98
5	1	0	1	3.306	783,788	32,369	0.780	3300.93
6	0	1	1	5.017	808,365	59,340	0.830	3155.68
7	0	1	−1	7.379	115,5783	62,326	0.851	3757.14
8	−1	0	−1	13.398	1,207,293	37,080	0.916	3336.44
9 #	0	0	0	6.042	973,375	62,039	0.832	3558.42
10	1	1	0	4.460	989,728	105,582	0.720	5103.66
11 #	0	0	0	6.042	973,375	62,039	0.846	3558.42
12	0	−1	−1	7.791	1,195,844	61,884	0.833	3822.58
13 #	0	0	0	6.054	975,458	62,138	0.839	4167.72
14	−1	−1	0	10.936	911,593	34,327	0.932	3158.42
15 #	0	0	0	6.031	977,418	62,089	0.833	3558.42
16	1	0	−1	4.957	1,203,094	99,607	0.740	3916.72
17 #	0	0	0	6.062	973,375	62,039	0.932	3558.42
Factors		Levels						
		Low (−1)		Middle (0)			High (+1)	
Mobile phase ratio (A:B)		25:75		18:82			10:90	
pH		1.5		1.7			1.9	
Flow rate mL/min		0.8		1			1.2	

# Center points of HPLC method.

**Table 3 molecules-28-00472-t003:** The results of ANOVA for multiple regression models.

Summary															
Source	Sequential *p* Value								Lack of Fit *p* Value						
	R1		R2		R3	R4		R5	R1		R2	R3	R4		R5
Linear	<0.0001		<0.0001		0.0026	<0.0001		0.0248	<0.0001		<0.0001	<0.0001	0.9806		0.1867
2FI	0.4811		0.6616		0.135	0.8889		0.264	<0.0001		<0.0001	<0.0001	0.9432		0.2032
Quadratic	<0.0001		0.0049		0.3969	0.8191		0.5808	<0.0001		0.0004	<0.0001	0.852		0.1314
Cubic	<0.0001		0.0004		<0.0001	0.852		0.1314							
Sequential Model Sum of squares (type-1)															
Source	Sum of squares								df						
Mean Vs total	791.28		1.63 × 10^13^		6.18 × 10^10^	12.14		2.20 × 10^8^	1		1	1	1		1
Linear Vs mean	106.21		3.48 × 10^11^		5.14 × 100^9^	0.0668		2.04 × 10^6^	3		3	3	3		3
2FI Vs Linear	2.3		8.85 × 10^8^		1.13 × 100^9^	0.0006		6.41 × 10^5^	3		3	3	3		3
Quadratic Vs 2FI	8.32		4.45 × 10^9^		5.28 × 100^8^	0.0011		3.20 × 10^5^	3		3	3	3		3
Cubic Vs Quadratic	0.3462		9.32 × 10^8^		1.08 × 100^9^	0.0014		7.69 × 10^5^	3		3	3	3		3
Residual	0.0006		1.32 × 10^7^		7.86 × 100^3^	0.0073		2.97 × 10^5^	4		4	4	4		4
Total	908.46		1.66 × 10^13^		6.97 × 10^10^	12.21		2.24 × 10^8^	17		17	17	17		17
Lack of fit test															
Linear	10.97		6.27 × 10^9^		2.74 × 10^9^	0.0032		1.73 × 10^6^	9		9	9	9		9
2FI	8.66		5.39 × 10^9^		1.61 × 10^9^	0.0026		1.09 × 10^6^	6		6	6	6		6
Quadratic	0.3462		9.32 × 10^8^		1.08 × 10^9^	0.0014		7.69 × 10^5^	3		3	3	3		3
Cubic	0		0		0	0		0	0		0	0	0		0
Pure error	0.0006		1.32 × 10^7^		7.86 × 10^3^	0.0073		2.97 × 10^5^	4		4	4	4		4
Model summary statistics															
Source	Standard deviation								R-squared						
Linear	0.9185		21,984.64		14,513.20	0.0283		394.79	0.9064		0.9822	0.6525	0.8648		0.5016
2FI	0.9308		23,233.85		12,683.67	0.0314		372.23	0.9261		0.9847	0.7958	0.8727		0.6592
Quadratic	0.2226		11,619.20		12,425.46	0.0352		390.14	0.997		0.9973	0.8628	0.8876		0.7379
Cubic	0.012		1815.16		44.33	0.0426		272.49	1		1	1	0.9059		0.9269
Fit summary and ANOVA using DOE models															
Summary															
Source	Adjusted R-squared					Predicted R-squared									
	R1	R2	R3	R4	R5	R1	R2	R3	R4	R5	R1	R2	R3	R4	R5
Linear	0.8848	0.9782	0.5723	0.8336	0.3866	0.8289	0.9631	0.2745	0.816	0.035					
2FI	0.8817	0.9756	0.6733	0.7964	0.4547	0.7232	0.9226	−0.0408	0.7576	−0.4254					
Quadratic	0.9932	0.9939	0.6865	0.7431	0.4009	0.9527	0.9578	−1.1945	0.5603	−2.1389					
Cubic	1	0.9999	1	0.6235	0.7078										
Sequential Model Sum of squares (type-1)															
Source	Mean square					F-value					*p*-value Prob > F				
Mean Vs total	791.28	1.63 × 10+^13^	6.18 × 10+^10^	12.14	2.2 × 10^8^										
Linear Vs mean	35.4	1.16 × 10+^11^	1.71 × 10^9^	0.0223	679,600	41.97	239.76	8.14	27.72	4.36	<0.0001	<0.0001	0.0026	<0.0001	0.0248
2FI Vs Linear	0.7676	2.95 × 10^8^	3.77 × 10^8^	0.0002	213,500	0.886	0.5465	2.34	0.2076	1.54	0.4811	0.6616	0.135	0.8889	0.264
Quadratic Vs 2FI	2.77	1.48 × 10^9^	1.76 × 10^8^	0.0004	106,700	55.96	10.99	1.14	0.3082	0.7009	<0.0001	0.0049	0.3969	0.8191	0.5808
Cubic Vs Quadratic	0.1154	3.11 × 10^8^	3.60 × 10^8^	0.0005	256,200	800.35	94.28	1.83 × 10^5^	0.2591	3.45	<0.0001	0.0004	<0.0001	0.852	0.1314
Residual	0.0001	3.30 × 10^6^	1.97 × 10^3^	0.0018	742,49.8										
Total	53.44	9.78 × 10+^11^	4.10 × 10^9^	0.7185	1.3 × 10^7^										
Lack of fit test															
Linear	1.22	6.97 × 10^8^	3.04 × 10^9^	0.0004	192,100	8450.09	211.45	154,800	0.1939	2.59	<0.0001	<0.0001	<0.0001	0.9806	0.1867
2FI	1.44	8.98 × 10^8^	2.68 × 10^8^	0.0004	181,400	10,013.39	272.4	136,400	0.2347	2.44	<0.0001	<0.0001	<0.0001	0.9432	0.2032
Quadratic	0.1154	3.11 × 10^8^	3.60 × 10^8^	0.0005	256,200	800.35	94.28	183,300	0.2591	3.45	<0.0001	0.0004	<0.0001	0.852	0.1314
Cubic															
Pure error	0.0001	1.32 × 10^7^	1965.20	0.0018	74,249.8										
Model summary statistics															
Source	Adjusted R-squared					Predicted R-squared					PRESS				
Linear	0.8848	0.9782	0.5723	0.8336	0.3866	0.8289	0.9631	0.2745	0.816	0.035	20.05	1.31 × 10^10^	5.72 × 10^9^	0.0142	3.92 × 10^6^
2FI	0.8817	0.9756	0.6733	0.7964	0.4547	0.7232	0.9226	−0.0408	0.7576	−0.4547	32.43	2.74 × 10^10^	8.20 × 10^9^	0.0187	5.79 × 10^6^
Quadratic	0.9932	0.9939	0.6865	0.7431	0.4009	0.9527	0.9578	−1.1945	0.5603	−2.1389	5.54	1.49 × 10^10^	1.73 × 10+^10^	0.034	1.28 × 10^7^
Cubic	1	0.9999	1	0.6235	0.7078										

**Table 4 molecules-28-00472-t004:** Represented accuracy data.

Levels	Concentration of Standard (µg/mL)	Concentration of Sample (µg/mL)	Mean Area ± SD (n = 6)	%RSD	% Recovery Mean Area ± SD	%RSD
LQC	4.8	6	598,171.3 ± 1850.81	0.309	109.7 ± 1.08	0.988
MQC	6	6	690,278.2 ± 5265.43	0.762	101.1 ± 0.76	0.761
HQC	7.2	6	843,344.6 ± 7804.51	0.925	101.9 ± 0.90	0.883

**Table 5 molecules-28-00472-t005:** Represented precision data.

Levels	Concentration (µg/mL)	Parameters									
		Interday Precision (Repeatability) (Mean Area ± SD) (n = 6)	%RSD	Inter Analyst (Mean Area ± SD) (n = 6)			%RSD	Intraday Precision (Mean Area ± SD) (n = 6)			%RSD
				Analyst 1	Analyst 2	Analyst 3		Day 1	Day 2	Day 3	
LQC	4.8	596,769.7 ± 4592.90	0.76	596,278.4 ± 3545.17			0.594	590,802.8 ± 5873.30			0.994
MQC	6	691,518.4 ± 4491.63	0.64	690,742.9 ± 4967.83			0.719	690,659.7 ± 5102.69			0.738
HQC	7.2	840,356.3 ± 9244.59	1.10	843,885.7 ± 7510.50			0.889	870,089.2 ± 6965.64			0.800

**Table 6 molecules-28-00472-t006:** Represented Robustness data of the method.

Variable	Value (mL/min)	Concentration (µg/mL)	(Mean Area ± SD) (n = 6)	%RSD	(Mean Rt ± SD) (n = 6)	%RSD
Flow rate (mL/min)	0.6	6	886,757.4 ±11,624.23	1.310	6.9 ± 0.02	0.403
	0.8	6	648,420.8 ± 5814.92	0.896	5.1 ± 0.03	0.756
	1	6	607,624.1 ± 2930.18	0.482	4.1 ± 0.05	1.212
Wavelength	365	6	613,765.6 ± 2492.74	0.400	5.1 ± 0.04	0.931
	370	6	690,351.9 ± 5983.36	0.931	5.1 ± 0.03	0.588
	375	6	706,587.3 ± 6579.47	0.931	5.1 ± 0.05	1.000

**Table 7 molecules-28-00472-t007:** Depicting system suitability parameters.

Parameter	Value	Limit
Tailing factor	0.991	<2
Theoretical plate	4446.667	>2000
HETP	31.7	Depends on theoretical plate
Peak purity index	0.999	>0.5

**Table 8 molecules-28-00472-t008:** A percentage entrapment efficiency study of XH in SLNs (n = 3).

% of XH Entrapped ± SD	% of XH Unentrapped ± SD	% Total Amount of XH Recovered ± SD	% RSD
75.31 ± 0.17	23.83 ± 0.07	99.13 ± 0.14	0.142
67.68 ± 0.62	33.21 ± 0.23	100.9 ± 0.50	0.496
71.5 ± 0.75	28.9 ± 0.15	100.4 ± 0.60	0.599

**Table 9 molecules-28-00472-t009:** Target method quality profile (TMQP) for HPLC method of XH.

QTPP Elements	Target	Justification
Target analyte	XH	HPLC method development for the detection of XH, active analyte in samples
Target sample	Active pharmaceutical ingredient (API), SLN	HPLC method development for XH is essential for their detection in API and SLN formulation
Sample form	API is in solid form, SLN dispersion	XH API is in solid form
Method type	RP-HPLC	In reverse phase chromatography, using a hydrophobic stationary phase offers the benefit of high retention for the majority of high lipophilic drug molecules. XH has high lipophilicity (log *p* 4.6). As a result, the reverse-phase technique would be more efficient.
Instrument requirement	Quaternary pump HPLC system with PDA Detector	Choosing a quaternary pump in comparison to the binary pump, this device delivers great accuracy in mixing mobile phase liquids, whereas PDA detector results in detection of degradants at various wavelengths.
Sample characteristic	Liquid sample	The analyte must be prepared in liquid form for quantification and detection by reverse phase chromatography because it must be miscible with the mobile phase.
Standard preparation	Standard dilutions of drug	For the separation of drug, the standard dilutions were prepared by using mobile phase mixture.
Sample preparation	Weighing, handling, sampling, mixing with solvents	To prepare samples accurate amount of drug should be weighed and mixed with the solvents in order to get stock. Then serial dilutions should be made after sonicating the stock.
Method application	For assay of XH	The developed method should be capable of assaying XH in bulk and in various nano lipid based carriers such as SLNs.

## Data Availability

The data presented in this study are available on request from the corresponding author.

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
