# Peer review of "Quality by Design (QbD) Based Method for Estimation of Xanthohumol in Bulk and Solid Lipid Nanoparticles and Validation"

_molecules, 2023, doi:10.3390/molecules28020472_

Round 1

Reviewer 1 Report

The study was well designed to develop analytical assay of RP-HPLC for estimation of Xanthohumol. This will be a good example case study for analytical science community. However, there are a few comments.

- Box-Benkin design was selected for optimization of the assay developed. BBD is a popular selection for optimization, but not the only method. It might be good to address why this method was selected over different methods such as CCC, CCF.

- Reference for AQbD is required. 

 is utilized for developing and validating 40 the simple, sensitive, cost effective reverse-phase high performance liquid chromatographic method 

Author Response

Respected Editor (Molecules Journal),

First of all, we would like to thank the editorial team for the opportunity and we are grateful for your valuable comments in this article. We would also like to express our sincere thanks to the learned reviewers for their sagacious suggestions. We went through the comments and added the required information. Please note that the changes are track change version.

If still any further action is required, then please let us know. We will be happy to address that also.

Reviewer #1

Comment 1 : Box-Benkin design was selected for optimization of the assay developed. BBD is a popular selection for optimization, but not the only method. It might be good to address why this method was selected over different methods such as CCC, CCF.

Response: First of all, I would like to thank the reviewer for providing the valuable comments to improve the strength of the manuscript. Suggestion has incorporated as per the comment (Lines: 118-125).

Comment 2: Reference for AQbD is required.

- is utilized for developing and validating 40 the simple, sensitive, cost effective reverse-phase high performance liquid chromatographic method

Response:  As per the suggestion, reference has incorporated in the manuscript (reference 9,10).

Reviewer 2 Report

The authors developed a robust HPLC assay to measure XH in the formulation by utilizing QBD approach. 

The author first provide a very thorough study design to understand variables which determine the performance of assay. Then applied the optimal parameters to separate XH in the solid nano particle, which sheds light to the potential of the method. 

The authors cover extensive details in the method development which is really helpful for audience who are in the similar field. 

I recommend to accept with the current version. 

Author Response

Respected Editor (Molecules Journal),

First of all, we would like to thank the editorial team for the opportunity and we are grateful for your valuable comments in this article. We would also like to express our sincere thanks to the learned reviewers for their sagacious suggestions. We went through the comments and added the required information. Please note that the changes are track change version.

If still any further action is required, then please let us know. We will be happy to address that also.

Reviewer #2

The author first provide a very thorough study design to understand variables which determine the performance of assay. Then applied the optimal parameters to separate XH in the solid nano particle, which sheds light to the potential of the method.

- The authors cover extensive details in the method development which is really helpful for audience who are in the similar field.

- I recommend to accept with the current version.

 Response: I would like to thank the worthy reviewer for the appreciation. No action taken.
